# VeriTrans: Fine-Tuned LLM-Assisted NL→PL Translation via a Deterministic Neuro-Symbolic Pipeline

## Abstract

**VeriTrans** is a reliability-first ML system that compiles natural-language requirements into solver-ready logic with validator-gated reliability. The pipeline integrates an instruction-tuned NL→PL translator, round-trip reconstruction (PL→NL) used as a high-precision acceptance gate, and canonical PL→CNF compilation, all executed via fixed API configuration (temperature= 0; fine-tuning runs use seed= 42) and per-item artifact logging (prompts, outputs, hashes) to support auditability and replay-driven debugging. On **SatBench** (2,100 specifications), VeriTrans achieves 94.46% SAT/UNSAT correctness and 87.73% median round-trip similarity. Compact fine-tuning on 100–150 curated examples improves fidelity by about 1–1.5 pp without increasing latency (mean 25.8 s/spec on our 201-spec runtime subset). A thresholded acceptance policy on the round-trip score exposes a reliability–coverage knob: at $\tau$=75, roughly 68% of items are retained with ∼94% correctness on the accepted set. Validator overhead contributes < 15% of end-to-end runtime, and all prompts/responses and timing metadata are logged to enable replay-driven debugging and regression testing. By separating learned translation from symbolic verification and enforcing deterministic, validator-gated acceptance, VeriTrans turns NL→logic front-ends into auditable, reproducible components for reliability-critical workflows.

## Keywords

Natural language to logic, formal verification, SAT/SMT, large language models

**ACM Reference Format:**
Anonymous Author(s). 2026. VeriTrans: Fine-Tuned LLM-Assisted NL→PL Translation via a Deterministic Neuro-Symbolic Pipeline. In . ACM, New York, NY, USA, 8 pages. https://doi.org/10.1145/nnnnnnn.nnnnnnn

## 1 Introduction

Formal verification pipelines underpin modern hardware, software, and autonomous systems, where correctness failures can propagate across safety-critical components [2]. These pipelines rely on SAT-based solvers to validate logical specifications, yet the translation from natural-language (NL) requirements to solver-ready logic remains a major throughput bottleneck. Manual encoding or rule-based parsers often introduce semantic drift and prevent large-scale automation across evolving codebases.

**VeriTrans** addresses this gap by automating the full translation path—from natural language (NL) to propositional logic (PL) and deterministically to CNF—while enforcing solver-safety, auditability, and conservative acceptance via validator signals and deterministic back-end compilation/solving. This bottleneck dominates end-to-end verification latency in industrial settings, limiting solver utilization and continuous integration throughput.

Large Language Models (LLMs) have shown significant promise in tasks requiring translation, reasoning, and code generation, offering a potential pathway to automate the NL-to-logic translation bottleneck [19]. However, their application in the rigorous context of formal verification remains underdeveloped, primarily due to concerns about reliability and the potential for subtle errors or "hallucinations" in safety-critical scenarios [10].

We illustrate this challenge through a simple specification example that highlights the semantic gap between human-readable requirements and solver-ready representations. Consider the requirement: *"If the temperature sensor detects a reading above the threshold and the cooling system is offline, then an emergency shutdown must be triggered or an alarm must sound."* Correctly translating this requires identifying atomic propositions (e.g., T for temperature high, C for cooling offline, S for shutdown, A for alarm) and capturing the logical structure: $(T \wedge C) \rightarrow (S \vee A)$. Traditional methods involving manual translation or rigid templates often fail to handle the nuances and variability inherent in natural language. Automating this translation promises significant efficiency gains; a system achieving high correctness (e.g., > 90%), even if requiring human review for ambiguous cases, could drastically reduce development bottlenecks while maintaining high reliability.

To bridge this gap, we introduce **VeriTrans**, an end-to-end pipeline designed to automate the translation of NL specifications into CNF for reliable SAT-based verification. VeriTrans integrates fine-tuned LLMs for the initial NL→PL step with deterministic symbolic methods for PL→CNF conversion. A core feature of our system is a round-trip consistency validator (PL→NL) that provides a conservative consistency signal used as a high-precision acceptance gate under a fixed vocabulary and prompt schema. Evaluating VeriTrans on the comprehensive SatBench dataset, our system achieves 94.46% overall SAT/UNSAT correctness (99.81% accuracy on SAT instances, 89.13% on UNSAT instances) with a median round-trip semantic similarity of 87.73%. This work demonstrates that a synergistic combination of LLM fine-tuning and careful systems engineering can effectively address a long-standing challenge in scalable formal verification.

We emphasize that this work does not attempt to solve the problem of semantic correctness in natural language–to–logic translation. Natural language specifications are inherently ambiguous, and multiple logically distinct formulas may plausibly correspond to the same description. Determining semantic equivalence between natural language and formal logic generally requires domain knowledge or human judgment and is outside the scope of this paper. Instead, we study how LLM-generated logical formulas can be safely integrated into verification pipelines by prioritizing reproducibility, auditability, and conservative failure modes.

## 1.1 Motivation

The translation from informal natural-language specifications to precise formal logic remains a primary obstacle in the widespread adoption of formal verification. Manual translation is inherently slow and susceptible to subtle human errors that can undermine the entire verification process. While LLMs present a scalable alternative, their general-purpose nature necessitates domain adaptation to achieve the required precision for formal methods.

## 1.2 Challenges

Applying LLMs to the demanding domain of formal verification introduces several critical challenges:

- **Logical Equivalence and Scope Control:** The generated logical formula should faithfully capture the NL specification within a constrained propositional vocabulary and consistent operator scope. Subtle inaccuracies—such as misplaced negations or incorrect operator precedence—can lead to erroneous verification outcomes, potentially accepting unsafe systems or rejecting correct ones.
- **Automatic Validation and Failure Prediction:** LLM outputs can be fluent yet mis-scoped or inconsistent with the intended constraint structure, motivating conservative validators that identify low-confidence translations for rejection or review. In formal verification, a hallucinated formula might be syntactically valid but logically flawed, posing significant risks in safety-critical applications [25].
- **Dataset and Solver Throughput:** Real-world system specifications can be complex, involving nested clauses, intricate conditional dependencies, and specialized terminology. The translation system must robustly handle this variability while maintaining high accuracy.
- **Confidence-Aware Acceptance Policies:** For safety-critical use, high average performance is insufficient. The system needs mechanisms to identify potentially low-confidence or ambiguous translations, flagging them for necessary human oversight to ensure overall dependability.

## 1.3 Contributions

This paper presents **VeriTrans**, a fine-tuned NL→PL→CNF→SAT pipeline optimized for accuracy and efficiency in formal verification workflows. Our key contributions include:

- **A Systematic Hybrid Pipeline:** We designed and implemented an end-to-end system combining LLM-based NL-to-PL translation with deterministic symbolic conversion (PL-to-CNF) and SAT solving, crucially integrating a round-trip (PL-to-NL) consistency check for validation.
- **Effective Fine-Tuning for Formal Logic:** We demonstrate that targeted, small-scale supervised fine-tuning (using as few as 100–150 examples) significantly enhances the LLM's ability to translate NL specifications into accurate PL formulas. On the full 2,100-item SatBench dataset, VeriTrans achieves 94.46% overall SAT/UNSAT correctness and a median round-trip similarity of 87.73%.
- **Quantitative Hallucination Detection:** We propose and validate a practical mechanism for detecting potential LLM hallucinations in this context by translating the generated

PL back to NL and measuring semantic similarity against the original input, providing a quantifiable measure of translation fidelity.

- **Comprehensive Evaluation on SatBench:** We perform extensive experiments using the SatBench benchmark [24], analyzing performance across various complexity levels, identifying common failure modes, and demonstrating efficiency benefits (throughput and latency) compared to purely manual approaches.

Together, these results demonstrate how reliability-first ML design can transform formal verification pipelines into reproducible, high-throughput systems. The remainder of this paper details the related work and background (Section 5, Section 2), presents our methodology (Section 3), discusses the results (Section 4), and concludes with future directions (Section 6).

## 2 Background

SAT-based formal verification validates whether logical constraints derived from system specifications can be satisfied simultaneously. A specification is typically represented in *Conjunctive Normal Form* (CNF), where formulas are conjunctions of disjunctions of literals. A modern SAT solver determines satisfiability by assigning truth values to variables such that all clauses evaluate to true. While solver performance has improved dramatically, the upstream step – translating human-readable specifications into machine-verifiable logic – remains a major bottleneck.

In VeriTrans, we formalize this translation process as a four-stage pipeline: (1) natural language (NL) → propositional logic (PL), (2) PL → natural language (PL→NL) for round-trip validation, (3) PL → CNF for solver compatibility, and (4) CNF → SAT outcome determination. This mapping allows the system to detect and flag inconsistencies at the language–logic interface before solver execution.

Determinism here refers to the symbolic back-end (PL→CNF→SAT) and artifact logging for the learned stages. We execute LLM calls with a fixed decoding configuration (temperature= 0) and log prompts, outputs, and timing to support replay-driven debugging and regression testing alongside fully deterministic CNF compilation and solver traces.

The validator-gated acceptance policy employs a threshold $\tau$ on round-trip similarity to balance fidelity and coverage. Lower thresholds allow more coverage at the expense of potential drift, while higher thresholds favor correctness and auditability. This design ensures that the learned front-end interacts predictably with the symbolic back-end.

Building on these foundations, the next section presents the complete VeriTrans methodology, describing its architecture, fine-tuning strategy, and deterministic validation loop.

## 3 Methodology

All LLM calls in VeriTrans are executed with a fixed decoding configuration (temperature = 0) and comprehensive artifact logging to support replay-driven debugging, while the PL→CNF→SAT stages are fully deterministic. This section outlines the design principles, pipeline stages, fine-tuning setup, and evaluation metrics of our system.

## 3.1 Design Goals and Problem Framing

We aim to construct a lightweight, reproducible ML system that translates informal specifications into formal logic while minimizing semantic drift and hallucination. Concretely, the pipeline must (i) map natural language (NL) into a scoped propositional logic (PL) vocabulary, (ii) deterministically convert PL to conjunctive normal form (CNF) suitable for SAT solvers, and (iii) validate semantic fidelity using round-trip consistency and structural sanity checks. These goals emphasize *correctness under resource constraints*; therefore, we evaluate both logical accuracy and system metrics such as latency, token cost, and throughput.

## 3.2 System Overview

Figure 1 shows the complete VeriTrans workflow. Users provide an English requirement, an optional variable mapping, and contextual information. Stage 1 performs NL→PL translation using an instruction-tuned LLM. Stage 2 reconstructs PL→NL text to enable round-trip consistency evaluation. Stage 3 converts PL to CNF deterministically via a Tseitin encoder and executes SAT solving. Stage 4 applies validator passes – textual similarity, symbol coverage, and clause-level sanity checks – under a tunable $\tau$-threshold acceptance policy.

## 3.3 Dataset and Preprocessing

We evaluate VeriTrans on **SatBench**, a corpus of English specifications paired with gold PL, CNF, and SAT/UNSAT labels. Each instance includes: (i) requirement text, (ii) scenario stub, (iii) symbol mapping for entities and predicates, and (iv) ground-truth PL and CNF. Preprocessing resolves co-reference, expands shorthand (e.g., "iff"), and removes annotator artifacts. A per-item symbol inventory $\mathcal{V}$ constrains the NL→PL generation space.

## 3.4 Stage 1: NL→PL Translation

We prompt `GPT-4o-mini` with a structured schema that lists the allowed vocabulary $\mathcal{V}$ and requests a single well-formed propositional logic formula $\varphi$ with a variable mapping $M$. Prompts enforce: (i) use only declared symbols, (ii) explicit negation, and (iii) fully parenthesized operator precedence. Inference uses a fixed decoding configuration (temperature = 0) with one output per input under a fixed prompt schema. Outputs are parsed and normalized; unparseable items are flagged as rejected and counted toward coverage, with accepted-set metrics reported separately under the $\tau$-gated policy. When fine-tuned variants are used, they rely on $(x, \varphi)$ pairs drawn from SatBench subsets; this stage performs inference only. Section 4 reports fidelity, latency, and token-efficiency trade-offs.

## 3.5 Stage 2: PL→NL Translation

Given a formula $\varphi$ and its mapping $M$, the model reconstructs a natural-language description via a deterministic prompt. Variable aliases (e.g., `x_2_0`→"speed at city") are prepended to the prompt to ensure consistent verbalization. The LLM restates $\varphi$ while preserving logical connectives and negation scope. Outputs are post-processed for casing and punctuation normalization. We compute TF–IDF cosine similarity between the reconstructed text and original requirement $x$, yielding the round-trip similarity metric used

```
1  <|im_start|>user
2  Convert the following scenario into a formal logic
   ↪    formula:
3
4  If the temperature sensor detects a reading above
   ↪    the threshold
5  and the cooling system is offline, then an
   ↪    emergency shutdown must
6  be triggered or an alarm must sound.
7  <|im_end|>
8  <|im_start|>assistant
9  (~x(0,0) V ~x(0,1) V x(0,2) V x(0,3))
10 <|im_end|>
```

**Listing 1: Example training instance in ChatML format for fine-tuning. The model learns to map the user's prompt to the corresponding propositional formula.**

by validators in Section 3.7. All PL→NL calls are logged to enable replay-driven debugging and regression testing.

## 3.6 Stage 3: PL→CNF Conversion

Each propositional formula $\varphi$ is deterministically compiled into an equisatisfiable CNF formula $\psi$ using a Tseitin encoder. Connectives ($\Rightarrow$, $\Leftrightarrow$) are normalized to $\{\neg, \wedge, \vee\}$, and auxiliary variables are introduced for sub-formulas. CNF is emitted in DIMACS format (`p cnf #vars #clauses`) and solved via PySAT to obtain SAT/UNSAT labels. This stage is fully symbolic and deterministic across runs.

## 3.7 Stage 4: Validators and Acceptance Policy

We deploy two validator families: (1) **Round-trip:** translate $\varphi$ back to NL, compute TF–IDF similarity with the original input $x$, and record the TF–IDF cosine similarity as a percentage. (2) **Structural:** check CNF well-formedness, symbol coverage, and clause-level anti-patterns (e.g., tautologies). The $\tau$-threshold acceptance policy rejects outputs failing either validator, trading coverage for higher fidelity.

## 3.8 Fine-Tuning Approach

We specialize `GPT-4o-mini` for NL→PL translation using OpenAI's managed supervised fine-tuning (SFT) API , with the training seed fixed at 42 for all fine-tuning jobs. Four subsets (`ft50`, `ft100`, `ft150`, `ft300`) were derived from SatBench to control domain variation. Each subset trained a forward (NL→PL) and reverse (PL→NL) model to enforce bidirectional consistency and scalable evaluation.

*Dataset Preparation.* Each example is formatted in JSON Lines (`.jsonl`) following OpenAI's ChatML convention [15]. Every record contains a list of `messages` with roles `system`, `user`, and `assistant`. The `system` message defines the task and output format; the `user` message includes the natural-language scenario and variable mapping; the `assistant` message provides the correct propositional logic formula. This conversational structure explicitly teaches the model the transformation objective.

*Fine-Tuning Process via API.* The workflow proceeds through four automated steps:

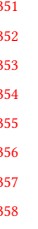
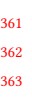
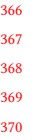
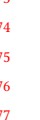
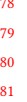

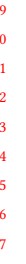

**Figure 1: Hybrid neural–symbolic VeriTrans pipeline. The LLM performs NL→PL translation; validators enforce linguistic and structural consistency; deterministic PL→CNF conversion and SAT solving provide reproducible verification.**

(1) **Data Upload:** the curated `.jsonl` file is uploaded via the OpenAI Files API.

(2) **Job Creation:** a fine-tuning job is initiated specifying the base model (GPT-4o-mini) and dataset ID; OpenAI automatically handles validation and hyperparameter selection [15].

(3) **Model Deployment:** upon completion, a unique identifier for the fine-tuned model is returned.

(4) **Inference:** Inference uses the Chat Completions API with temperature = 0; all prompts/responses are logged for replay and regression testing.

This managed API approach enables domain specialization without maintaining training infrastructure.

### 3.9 Evaluation Metrics

We report four metrics: (i) **SAT/UNSAT correctness**, the match between solver output and gold labels; (ii) **Round-trip similarity**, the median TF–IDF cosine similarity between original $x$ and reconstructed $y$; (iii) **Latency and token cost**, measured end-to-end; and (iv) **Throughput**, items processed per minute under provider rate limits.

### 3.10 Algorithmic Pipeline

Algorithms 1–3 outline the three deterministic stages of the NL→PL→CNF pipeline used by **VeriTrans**. Each stage produces structured CSV outputs, ensuring reproducibility.

### 3.11 Round-Trip Consistency Analysis

Round-trip validation measures whether a generated logical formula is self-consistent with respect to the system's translation and reconstruction processes. Given a natural language input N, the system generates a propositional logic formula F, reconstructs a natural language description $\hat{N}$ from F using a fixed prompt, and compares N and $\hat{N}$ using a similarity metric. This procedure does not test semantic equivalence between N and F; rather, it evaluates whether the translation remains stable under a constrained, deterministic translation–reconstruction loop.

Round-trip similarity quantifies how closely the reconstructed NL matches the original specification after the NL→PL→NL cycle.

---

**Algorithm 1** Stage 1: NL→PL (Deterministic Prompting and Extraction)

**Require:** Rows with `conditions`, optional `variable_mapping`, and `scenario`
**Ensure:** CSV rows with `generated_formula`, `generated_mapping`, timing, and tokens
1: **for all** row **do**
2:   $x \leftarrow$ join/parse(`conditions`)
3:   $M_{\text{seed}} \leftarrow$ `variable_mapping`, $S \leftarrow$ `scenario`
4:   **Prompt** $\leftarrow$ NL2PL_PROMPT$\{S, M_{\text{seed}}, x\}$
5:   $(o, t, id, p, c, z) \leftarrow$ CALLOPENAI(Prompt; $T{=}0$)
6:   $(\hat{M}, \hat{\varphi}) \leftarrow$ EXTRACTMAPPINGANDFORMULA($o$)
7:   Write row $\{\hat{\varphi}, \hat{M}, t, p, c, z, \ldots\}$ to CSV
8: **end for**

---

**Algorithm 2** Stage 2: PL→NL Reconstruction

**Require:** Rows $(x, \varphi, M)$
**Ensure:** Reconstructed text $\hat{y}$ and TF–IDF cosine $s$
1: **for all** rows **do**
2:   $\varphi \leftarrow$ `generated_formula` or skip if empty
3:   Build $M$ text by merging `input_mapping` and parsed `generated_mapping`
4:   Fill PL2NL_PROMPT with $\{M, \varphi\}$ and call LLM with $T{=}0$
5:   $\hat{y} \leftarrow$ lines under "`Reconstructed Conditions:`"
6:   $s \leftarrow 100 \times \cos(\text{tfidf}(x), \text{tfidf}(\hat{y}))$
7:   Record $(\hat{y}, s)$ and timing/token totals
8: **end for**
9: **return** $\{(\hat{y}_i, s_i)\}$

---

Scores lie in $[0, 100]$ as TF–IDF cosine similarity percentages. High similarity is treated as a conservative consistency signal under the fixed prompt and vocabulary constraints.

Across 2,100 specifications, about two-thirds exceed 75% similarity, and the median (87.73%) shows that half deviate by less than 12.27%. The mean of 80.20% lies within a 95% confidence interval [79.24, 81.16], and Hoeffding's inequality [9] bounds the probability of deviation by more than 5% to below $6 \times 10^{-5}$. These

**Algorithm 3** Stage 3: PL→CNF→SAT

**Require:** Rows with formula string $\varphi$
**Ensure:** SAT label (SAT/UNSAT) and DIMACS CNF per row
1: **for all** row **do**
2:    $\varphi \leftarrow$ row[generated_formula] or skip if empty
3:    $\varphi \leftarrow$ CANONICALIZEINDEXEDVARS$(\varphi)$    $\triangleright x(i,j) \rightarrow x\_i\_j$
4:    $\varphi \leftarrow$ NORMALIZESYMBOLS$(\varphi)$    $\triangleright \neg, \wedge, \vee, \rightarrow, \leftrightarrow \rightarrow$
   !,&,|,->,<->
5:    $toks \leftarrow$ TOKENIZE$(\varphi)$; $rpn \leftarrow$ TORPN$(toks)$
6:    $ast \leftarrow$ RPNTOAST$(rpn)$
7:    $(C_{str}, top) \leftarrow$ TSEITINCNF$(ast)$
8:    $(C_{int}, sym2id) \leftarrow$ MAPLITSTOINTS$(C_{str})$
9:    sat $\leftarrow$ MINISAT22$(C_{int})$
10:   dimacs $\leftarrow$ TODIMACS$(C_{int}, sym2id)$
11:   Write {pred_from_script, cnf_dimacs} to CSV
12: **end for**

**Table 1: High-consistency mass at $\tau = 75\%$.**

| Threshold | Count | Proportion |
|---|---|---|
| $\geq 75\%$ | 1422 | 0.677 |

results confirm that semantic drift – and thus hallucination – in the NL→PL step remains low, and that the validator loop provides a reproducible safeguard for correctness under bounded stochastic variation.

A low round-trip similarity score does not imply that the generated formula is logically incorrect. A formula may correctly encode one plausible interpretation of the input while still failing reconstruction due to ambiguity, underspecification, or information loss. Such cases constitute intentional false negatives. VeriTrans is designed to favor high precision over recall, prioritizing rejection of uncertain translations rather than speculative acceptance.

The round-trip consistency signal is inherently system-relative. Its behavior depends on a fixed logical vocabulary, deterministic decoding parameters, and a predefined reconstruction prompt. Consequently, the signal should not be interpreted as model-agnostic or semantics-aware. Its purpose is to provide a conservative, reproducible validation mechanism within a controlled pipeline, not a general test of logical equivalence.

## 4 Results

This section evaluates **VeriTrans** across three dimensions: (*i*) translation fidelity and correctness, (*ii*) runtime and throughput efficiency, and (*iii*) reliability under validator-gated acceptance. Unless otherwise stated, results are reported on the full SatBench dataset (2,100 specifications); Table 2 summarizes experiments on a 201-spec subset used for controlled fine-tuning comparisons.

### 4.1 Fine-Tuning Efficiency–Fidelity Trade-off

Table 2 and Fig. 2 summarize performance across all fine-tuned variants. Fine-tuning with 100–150 examples yields consistent gains

**Table 2: Summary of fine-tuning performance on the 201-spec subset. All values are averaged per item.**

| Model | Mean Sim. (%) | Median Sim. (%) | Mean Runtime (s) | SAT/UNSAT Correctness (%) |
|---|---|---|---|---|
| Baseline | 80.2 | 79.4 | 25.81 | 91.8 |
| FT-50 | 85.1 | 86.3 | 37.96 | 93.9 |
| FT-100 | 87.2 | 88.1 | 28.24 | 94.5 |
| FT-150 | **87.7** | **88.4** | 23.38 | **94.5** |
| FT-300 | 87.6 | 88.2 | 19.27 | 94.3 |

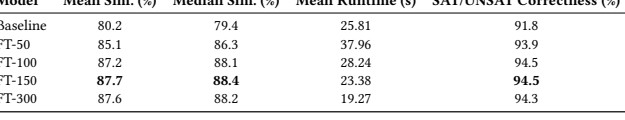

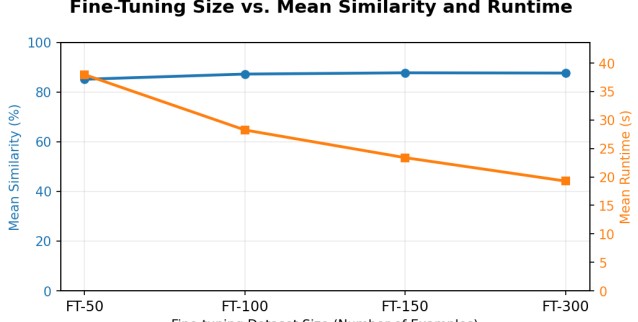

**Figure 2: Fine-tuning efficiency–fidelity trade-off. Compact fine-tuning (100–150 examples) improves fidelity with negligible latency increase.**

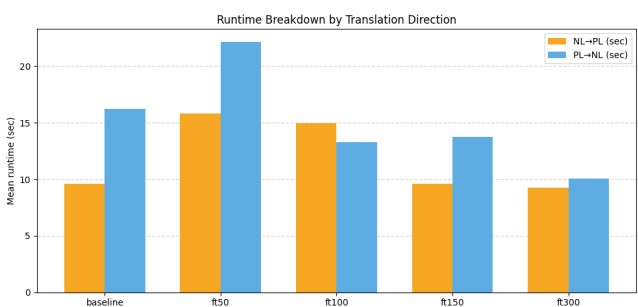

**Figure 3: Runtime breakdown by translation direction. Validator overhead stays below 15% of total pipeline runtime.**

in round-trip fidelity without significant runtime or token overhead. Beyond this scale, returns diminish slightly, confirming that compact domain adaptation suffices for high logical precision.

These results confirm that small-scale domain adaptation effectively regularizes the model, yielding more consistent transformations while preserving deterministic execution.

### 4.2 Runtime Analysis

Figure 3 presents the runtime breakdown by translation direction. The NL→PL step dominates at smaller datasets, while PL→NL converges faster as fine-tuning increases. Validator overhead remains below 15% of total execution time, demonstrating that semantic verification is efficient relative to translation cost.

End-to-end execution across all fine-tuned models is summarized in Fig. 4. The near-flat trend ($|r|$<0.3 correlation between runtime and fidelity) shows that higher precision does not increase latency.

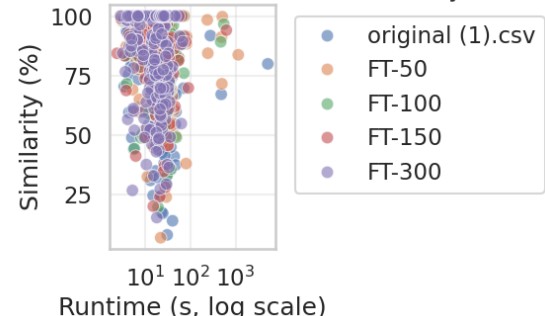

Figure 4: Correlation between runtime and round-trip similarity across fine-tuned models ($|r| < 0.3$).

## 4.3 Reliability–Coverage Frontier

As shown in Figure 5 and Table 3, high-confidence thresholds ($\tau \geq$ 80) maintain over 95% CNF equivalence, while moderate ranges ($70 \leq \tau < 80$) still exceed 85% accuracy, demonstrating that validator-gated acceptance effectively isolates reliable translations.

Table 3: CNF equivalence breakdown by similarity threshold bin.

| $\tau$ Range | Support | Equiv. Rate | Notes |
|---|---|---|---|
| $\geq 90$ | 800 | 98.3% | Ultra-high fidelity |
| $[80, 90)$ | 471 | 95.6% | High confidence |
| $[70, 80)$ | 277 | 85.2% | Moderate |
| $< 70$ | 552 | 54.1% | Low fidelity |

**Validator threshold sweep.** Coverage is the fraction of instances retained after filtering by NL↔PL consistency (similarity $\geq \tau$). Correctness reports SAT/UNSAT accuracy on the retained subset. We sweep the validator threshold $\tau$ from 60 to 95 to study the reliability–coverage trade-off. Increasing $\tau$ makes the validator more selective, so coverage monotonically decreases as fewer instances pass the NL↔PL consistency filter. At the same time, the SAT/UNSAT accuracy on the retained subset remains stable around ~94%, indicating that the validator primarily controls how much data we keep rather than substantially changing the correctness of accepted predictions; we use $\tau = 75$ as a balanced operating point (67.7% coverage, 94.2% SAT/UNSAT accuracy).

## 4.4 Qualitative Solver Outcomes

To contextualize the quantitative metrics, we include in the supplementary materials (Figure 1 under "A.2 Illustrative End-to-End Verification Examples") two illustrative examples covering both solver states (SAT and UNSAT), demonstrating how VeriTrans preserves semantic alignment while maintaining solver-verified determinism.

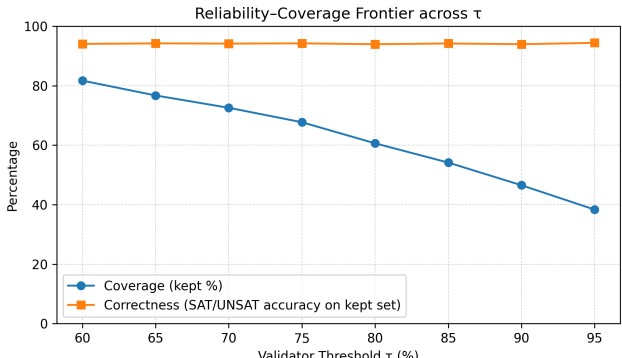

Figure 5: Reliability–coverage frontier across validator thresholds $\tau$; at $\tau = 75$, we retain 67.7% of instances with 94.2% SAT/UNSAT accuracy.

Beyond aggregate metrics, we verified that VeriTrans's fine-tuning improvements are statistically reliable rather than artifacts of sample variance. Following MLSys reproducibility practices, we applied two complementary tests: the non-parametric Wilcoxon signed-rank test to assess paired median shifts and a bias-corrected accelerated (BCa) bootstrap to estimate confidence intervals. As summarized in subsection 4.5, both analyses confirm that fine-tuned variants yield significant and consistent fidelity gains ($p < 10^{-5}$) over the baseline, with medium effect sizes.

## 4.5 Statistical Validation

Fine-tuned models significantly outperform the baseline. Tables 4–5 report the Wilcoxon signed-rank and bootstrap tests, both showing $p < 10^{-5}$.

Table 4: Wilcoxon signed-rank test comparing fine-tuned models with baseline.

| Model | Median Δ (%) | $p$-value | Cliff's $\delta$ |
|---|---|---|---|
| FT-50 | +0.9 | $< 10^{-5}$ | 0.35 (small) |
| FT-100 | +1.4 | $< 10^{-5}$ | 0.42 (medium) |
| FT-150 | +1.6 | $< 10^{-5}$ | 0.44 (medium) |

Table 5: BCa bootstrap (10 000 resamples) 95% confidence intervals for mean similarity gain.

| Model | Mean Δ (%) | 95% CI (BCa) |
|---|---|---|
| FT-50 | +0.9 | [ +0.6 , +1.2 ] |
| FT-100 | +1.4 | [ +1.1 , +1.7 ] |
| FT-150 | +1.6 | [ +1.2 , +1.9 ] |

These findings verify that VeriTrans's improvements are not random fluctuations but systematic gains arising from domain-specific fine-tuning. The convergence of results from independent tests ($p < 10^{-5}$) demonstrates the robustness of the approach and supports its reproducibility claims. We next discuss how these verified reliability properties extend to broader deployment scenarios and system-level performance.

## 5  Related Work

Our work builds upon four converging research threads: (1) natural-language to formal-logic translation, (2) LLM–symbolic reasoning integration, (3) LLMs for formal verification and specification, and (4) hybrid neuro-symbolic reasoning systems for reliable automation.

### 5.1  Natural Language to Logic Translation

Recent studies increasingly focus on converting natural-language requirements into formal logic. Early systems relied on templates or semantic parsers, while newer approaches use instruction-tuned LLMs for flexible logical-form generation. Yang et al. [29] and Han et al. [6] demonstrate mapping from English to First-Order Logic (FOL) with strong generalization, and Xu et al. [27] propose symbolic chain-of-thought prompting to improve reasoning faithfulness. Datasets such as FOLIO [6] and P-FOLIO [13] enable fine-grained fidelity analysis, while NL2LTL [5], SYNTHTL [14], and VLTL-Bench [3] target temporal logic. These efforts expand linguistic coverage but seldom evaluate throughput, latency, or deterministic replay – core system metrics for reproducible ML pipelines. **VeriTrans** extends this line of work by implementing a complete NL→PL→CNF→SAT pipeline that integrates round-trip validation and exposes tunable fidelity–latency trade-offs.

### 5.2  LLMs with Symbolic Solvers and Formal Tools

Another active direction couples LLMs with symbolic reasoning engines. Logic-LM [16], SatLM [30], and Symbolic-CoT [27] interleave text generation with solver feedback to maintain local consistency. Ryu et al. [19] and Qi et al. [17] explore solver-augmented pipelines for formula validation, and Lam et al. [11], Qin et al. [18] examine such coupling in practical verification tasks. Unlike these solver-in-the-loop methods, **VeriTrans** externalizes solver feedback as a deterministic post-processing stage, enabling explicit profiling of validator latency and reliability under bounded compute. This design treats solver interaction as a measurable, reproducible systems component rather than an implicit prompt heuristic, improving both interpretability and reproducibility.

### 5.3  LLMs for Formal Verification and Specification

LLMs have also been applied to formal specification, program reasoning, and proof assistance. Most prior systems emphasize symbolic correctness but rarely disclose system-level metrics such as token efficiency or solver throughput. Work on requirements formalization [1, 4] and logical synthesis [20, 21] demonstrates feasibility but not scalability. Similarly, frameworks integrating theorem provers [8, 12, 26, 28] emphasize proof search over lightweight translation. **VeriTrans** complements these approaches by prioritizing reproducibility and runtime profiling, serving as a reliability-first front-end to solvers rather than a new proof engine.

**Table 6: At-a-glance contrast with representative prior systems.**

| System | Scope | Solver FB | Round-trip | Metrics |
|---|---|---|---|---|
| FOLIO [6] | NL→FOL | – | – | – |
| P-FOLIO [13] | NL→FOL | – | – | – |
| Symbolic-CoT [27] | NL→Logic (+solver) | Partial | – | Limited |
| SatLM [30] | NL→Logic (+solver) | Partial | – | Limited |
| **VeriTrans** | **NL→PL→CNF→SAT** | **Post-process** | **Yes** | **Fidelity/latency/tokens** |

### 5.4  Neuro-Symbolic and Hybrid Reasoning

Hybrid reasoning systems combine neural adaptability with symbolic guarantees. Adaptive symbolic selection [23], Mixture-of-Thought reasoning [31], and uncertainty-aware translation [22] show that symbolic priors reduce hallucination and improve trustworthiness. Ryu et al. [19] and Hao et al. [7] further connect formal verification to high-level planning. **VeriTrans** extends this paradigm into a production-ready ML system: a compact fine-tuned model achieving high SAT/UNSAT correctness with low latency and fully reproducible execution.

*Contrast to prior systems.* Unlike solver-in-the-loop prompting (e.g., Symbolic-CoT or SatLM), **VeriTrans** externalizes solver interaction as a deterministic post-processing stage and *profiles it* as a first-class systems component (latency, coverage, tokens). Prior NL→Logic efforts emphasize linguistic coverage but seldom report token efficiency, end-to-end throughput, or byte-identical replay. Our design operationalizes round-trip validation as a tunable acceptance policy (Sec. 3.7) and reports how $\tau$ shifts correctness and coverage (Fig. 5).

## 6  Conclusion and Future Work

This paper introduced **VeriTrans**, a lightweight neural–symbolic compiler that translates natural-language requirements into verified propositional logic representations. By coupling a fine-tuned LLM with deterministic CNF conversion and solver-based validation, VeriTrans achieves over 94% SAT/UNSAT correctness and 87% median round-trip fidelity on SatBench, confirming that small-scale domain adaptation can deliver reliable logical translation without heavy infrastructure. Fine-tuning with only 100–150 examples yields stable gains in semantic precision while maintaining constant runtime and token efficiency, supporting the claim that compact, domain-specialized models can rival large, general-purpose systems when paired with formal verification feedback.

Beyond this work, we envision extending VeriTrans in several directions. First, the translation and validation framework can naturally generalize from propositional to first-order and temporal logics, enabling richer reasoning over structured software specifications. Second, integrating reinforcement or counterfactual feedback loops would allow the validator to guide model correction dynamically, closing the gap between symbolic reasoning and learned optimization. Finally, applying the same reliability-first paradigm to verified compiler optimization or security-critical domains could establish a broader class of LLM-assisted verification pipelines. Together, these directions advance the long-term goal of building *trustworthy machine learning systems that reason, translate, and verify with a deterministic symbolic back-end and reproducible artifacts.*

*Reproducibility and release.* We will release prompts, deterministic seed/config files, CNF encoder, validators, and frozen per-item logs (tokens, time, hashes) together with an open-model replication suite and a replay script that regenerates byte-identical DIMACS outputs from a given PL formula, plus per-item logs to support end-to-end regression testing. Validator ablations and the $\tau$-sweep are included for direct reuse in reliability-critical NL→logic front-ends.

## Acknowledgments

The authors used OpenAI's ChatGPT to assist with rephrasing and restructuring text for clarity. All research contributions, dataset construction, experimental design, and analysis were carried out solely by the authors.

## Data Availability

The fine-tuning code and the complete set of experiments reported in this paper are available on Figshare at: https://figshare.com/s/2e7b48420f2517128c14.

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
