# OpenReview forum: "VeriTrans: Fine-Tuned LLM-Assisted NL→PL Translation via a Deterministic Neuro-Symbolic Pipeline"
_ACM.org/AIWare/2026/Conference — AIware 2026_

### Official Review · Reviewer_VQrH · 2026-03-08

**Rating:** 3
**Confidence:** 4

**Review:**

Strengths


* The paper is very well-written and easy to follow. The pipeline is well motivated, and the decomposition into LLM-based NL → PL translation, followed by deterministic symbolic back-end is sensible for verification-oriented use cases. This design also gives the system a good balance between flexibility and auditability.
* The paper includes fairly comprehensive ablations and system-level evaluation. In particular, it studies reliability–coverage tradeoffs under different validator thresholds, latency, and the effect of small-scale fine-tuning.
* The round-trip design (NL→PL→NL) is a natural choice for a practical consistency check, even though I have concerns about the specific scoring mechanism used.


Weaknesses


* The main evaluation metric, SAT/UNSAT correctness, is too coarse to validate overall translation quality. Matching the final satisfiability label only shows that the pipeline often reaches the expected end decision. It does not establish that the intermediate NL → PL translation is semantically faithful, as incorrect formulas can still produce the same SAT/UNSAT outcome.
* Related, the NL → PL → NL round trip is a plausible way to check the NL → PL translation. However, the current consistency check seems to rely mostly on lexical and syntactic similarity rather than semantic equivalence. As a result, it is unclear how well this validator can separate truly faithful translations from outputs that are syntactically similar but differ in semantics.


Questions for author

* The reported SAT/UNSAT correctness is encouraging as an end-task metric, but do the authors consider any more direct evidence of formula-level semantic fidelity beyond satisfiability-label agreement?
* The NL → PL → NL round trip design is interesting, but the current validator appears to rely mainly on textual similarity. Did the authors explore more semantics aware validation signals beyond TF IDF based similarity?
* The fine tuning gains appear relatively modest. Could the authors comment on the practical significance of these improvements?


Overall, I think this is a well written paper with a practical reliability oriented framing. I also appreciate that the paper evaluates the system from multiple angles. My main concern is that the current evaluation still falls short of validating whether the NL → PL translation is semantically correct. In particular, SAT/UNSAT correctness is a coarse end task metric, and the round trip validator appears too syntactic to provide strong evidence of semantic fidelity.

**Summary:**

This paper presents VeriTrans, a reliability-oriented pipeline for translating natural language (NL) specifications into propositional logic (PL), then deterministically converting them to conjunctive normal form (CNF) and solving them with a SAT solver. The system uses a fine-tuned GPT-4o-mini model for NL → PL, a reverse PL → NL round-trip reconstruction as a validator, and deterministic PL → CNF → SAT back-end stages with logging for auditability and replay. On SatBench, the paper reports 94.46% SAT/UNSAT correctness, 87.73% median round-trip similarity, and a reliability–coverage tradeoff controlled by a round-trip threshold.

---

> ### Author Response · Authors · 2026-03-15
> **Response on evaluation fidelity, validator design, and practical significance**
>
> Thank you for the positive review and for carefully engaging with both the strengths and the current limits of the evaluation.
> We agree that SAT/UNSAT correctness is a coarse end-task metric rather than proof of full formula-level semantic fidelity. For that reason, the paper does not rely solely on SAT/UNSAT correctness; it also includes round-trip validation, structural checks, and reliability-coverage analysis. Our intention is not to suggest that end-task agreement by itself establishes semantic faithfulness.
> We also agree that the round-trip validator relies on textual similarity and is not semantics-aware. This is an intentional design choice rather than an implicit assumption. In VeriTrans, the validator is meant to be a conservative gate under fixed vocabulary and deterministic prompts, not a semantics oracle. More precisely, it serves as a lightweight, deterministic acceptance heuristic to detect reconstruction drift within the controlled pipeline. This design follows directly from the paper’s reliability-first framing and emphasis on auditability and replayability.
> We agree that more semantics-aware validation signals would be valuable. In the current paper, however, we prioritized deterministic replayability and auditability over introducing an additional learned evaluator. We view that as a deliberate trade-off aligned with the paper’s system scope.
> On the practical significance of the fine-tuning gains, our intended claim is modest. The key point is that compact fine-tuning improves fidelity while preserving comparable latency and keeping validator overhead low. In a reliability-oriented pipeline, that kind of stable gain is meaningful even if it is not dramatic in absolute size, because it strengthens consistency without compromising reproducibility or efficiency.
> Thank you again for the constructive feedback and encouraging assessment.

---

### Official Review · Reviewer_jMtF · 2026-03-09

**Rating:** 2
**Confidence:** 4

**Review:**

Strengths
- The problem (NL -> PL translation) is interesting and useful
- The paper addresses hallucination detection, which is a challenging problem
- The approach is evaluated under several configurations

Weakness
- The writing quality could be improved in certain sections (see below)
- The overall approach is fairly simple, and is not compared to the original results in the SatBench paper
- Using cosine similarity in the evaluation to judge the equivalence between the original requirements and the PL -> NL requirements is unsound (see below)

Writing Quality Improvements
- The introduction could flow better. VeriTrans is introduced three times and the results are presented twice throughout section 1. I recommend structuring the flow as motivation, challenges, limitations of prior work, and then introduce veritrans, and then present results.
- The introduction also lacks discussion about related work. The contributions are not positioned relative to related work so it is difficult to evaluate the actual contribution of the work.
- The results section was difficult to follow. It should first present a set of research questions which can be answered with a Yes/No, or a specific metric. Then the remainder of the section should present the answers.
- It’s unclear how SatBench is used. The SatBench paper states that it contains logic puzzles and questions, whereas the VeriTrans paper states it contains requirements in section 3.3.

Use of Cosine Similarity in Evaluation
- Using cosine similarity in the experimental evaluation to assess whether the original and back-translated requirements has issues. It’s unclear what cosine similarity is actually flagging as “not equivalent” when it computes a low score. Prompting an LLM to judge equivalence would likely provide less ambiguous judgements (though it would still not be perfect)
- I think what the authors really want to measure is the uncertainty or ambiguity in the NL requirements (which leads to incorrect translations). There is a large body of work on measuring uncertainty of LLMs that the authors can draw on, and could be discussed.

Questions for the authors:
- Could you provide more details about how SatBench was used/adapted to evaluate VeriTrans? From my cursory reading of SatBench, the benchmark consists of logic puzzles, rather than requirements as stated in Section 3.3.
- The SatBench paper also evaluates an approach on the benchmark itself. Could you compare your evaluation to the evaluation in the original SatBench paper?

**Summary:**

The authors propose a pipeline for translating NL requirements to formal logic for the purpose of using symbolic reasoning to find inconsistencies in the NL requirements. To validate the correctness of the translation, the authors propose to back-translate the formal logic to NL requirements and compare those to the original requirements using a similarity measure. The authors show that their pipeline performs well on SatBench, a benchmark containing natural language logic puzzles which have a corresponding SAT/UNSAT result.

---

> ### Author Response · Authors · 2026-03-15
> **Response on SatBench framing, evaluation scope, and presentation**
>
> Thank you for the detailed review and for highlighting several areas where the paper’s presentation can be improved.
> We agree that the core pipeline is intentionally simple. This is by design. The contribution is not a complex prompting strategy, but a reliability-oriented decomposition that separates learned NL→PL translation from deterministic PL→CNF→SAT execution, then adds validator-gated acceptance and replayable artifact logging. The intended contribution is therefore system-level reliability and reproducibility, rather than novelty through prompt complexity alone.
> We also agree that the introduction and results sections could be structured more clearly. In a revision, we would streamline the introduction to make the progression from motivation and challenges to scope, system overview, and key findings more direct. We also agree that an explicit framing of the research question would make the results section easier to follow.
> On SatBench, we believe the main issue is one of wording rather than methodology. We use SatBench as a benchmark of English descriptions paired with gold PL, CNF, and SAT/UNSAT labels. In our study, those English descriptions are treated as structured natural-language, specification-like inputs whose logical constraints are to be recovered. We are not claiming that SatBench is an industrial requirements corpus. We agree that the paper should have made this distinction clearer and used more precise wording than simply “requirements” throughout.
> On comparison with the original SatBench paper, this is a reasonable request. Our use of SatBench differs from the original benchmark objective because we evaluate a deterministic pipeline with validator-gated acceptance, latency, and coverage trade-offs, and replayability. For that reason, our evaluation target is not identical to the original benchmark framing. We agree that this difference should have been explained more explicitly.
> Regarding cosine similarity, we agree with the underlying concern and appreciate the opportunity to clarify the intended claim. The paper does not use cosine similarity as a certificate of equivalence. It is intended only as a conservative consistency signal within a constrained deterministic loop. More precisely, it is used as a lightweight heuristic for reconstruction drift, not as a semantics-aware validation claim.
> Thank you again for the careful reading and helpful comments.

---

### Official Review · Reviewer_LDpM · 2026-03-13

**Rating:** 3
**Confidence:** 2

**Review:**

# Strengths
- Identify a realistic problem in safety-critical formal verification pipelines.
- Provide a verifiable, deterministic approach that safely integrates LLMs with symbolic solvers.
- The proposed round-trip validation serves as an effective, high-precision acceptance gate.
- A replication package and frozen per-item logs are promised to be provided.

# Weaknesses
- Missing justifications for relying purely on TF-IDF as the similarity metric.
- Missing direct quantitative comparisons with state-of-the-art baselines.
- Experiments are limited strictly to Propositional Logic.
- Limitative qualitative error analysis.

# Detailed comments for authors
## Importance
The work addresses an important problem of translating informal natural-language specifications to precise formal logic. This is a major throughput bottleneck in industrial formal verification. The work addresses this important direction by prioritizing reproducibility and conservative failure modes over pure generative flexibility.

## Originality
The work proposes a novel round-trip consistency validator (PL-to-NL) that provides a conservative consistency signal. Enforcing deterministic configurations (temperature=0, seed=42) alongside symbolic compilation is a pragmatic and original systems-engineering approach to integrating LLMs into verification.

## Soundness
The proposed approach and its evaluation seem generally sound. Evaluation is done using the SatBench dataset, which includes English specifications paired with gold PL, CNF, and SAT/UNSAT labels. The inclusion of rigorous statistical validation, such as the Wilcoxon signed-rank test and BCa bootstrap, is highly appreciated.

However, the evaluation relies on TF-IDF cosine similarity between the reconstructed text and the original requirement for the round-trip validation. There are critical limitations to using TF-IDF for logical requirements. A single missing or added negation (e.g., "not") will barely change a TF-IDF score but will completely invert the logical satisfiability. The paper needs to better justify why a semantic metric (like BERTScore) or an LLM-as-a-judge approach was not used, or at least provide an analysis showing TF-IDF's false positive rate for scoping and negation errors.

The evaluation includes an ablation on fine-tuning sizes (FT-50 to FT-300) compared against a baseline. However, it lacks a direct comparison against other existing LLM-based formal translation systems. It would have been better if quantitative comparisons against systems mentioned in the related work, such as SatLM or Symbolic-CoT, were included to benchmark the pipeline's relative effectiveness.

The experiments can also be strengthened by discussing other programming or system logics. Currently, only Propositional Logic (PL) is investigated. Real-world software and hardware specifications often require First-Order Logic (FOL) or Linear Temporal Logic (LTL).

The paper would also have been stronger if a qualitative study highlighting concretely why ~5.5% of cases failed the SAT/UNSAT correctness che conducted. Most of the experiment results are rather quantitative, and understanding the failure modes (e.g., variable mapping errors vs. nested clause hallucinations) would be highly valuable.

# Verifiability and Transparency
The paper includes sufficient information and states that a replication package with prompts, deterministic seed files, and frozen logs will be released.

# Presentation
The paper is reasonably well written.

# Questions
1. How robust is the TF-IDF metric to small lexical changes that cause major logical shifts (e.g., missing negations), and why were semantic metrics not used?
2. Why are there no direct quantitative comparisons with other existing baselines (like SatLM or Symbolic-CoT) on the SatBench dataset?
3. Could you perform a qualitative error analysis on the instances where the SAT/UNSAT correctness failed as described in my review?

**Summary:**

This paper addresses the bottleneck of translating natural language specifications into solver-ready logic for formal verification. To address this problem, the authors propose VeriTrans, a deterministic neuro-symbolic pipeline. The pipeline integrates an instruction-tuned LLM for NL-to-PL translation, a round-trip (PL-to-NL) validator, and deterministic compilation to CNF for SAT solving. The experiments show that on the SatBench dataset (2,100 specifications), the system achieves 94.46% SAT/UNSAT correctness and an 87.73% median round-trip similarity. They also demonstrate that compact fine-tuning on 100-150 examples improves fidelity without increasing latency.

---

> ### Author Response · Authors · 2026-03-15
> **Response on validator design, baselines, and scope**
>
> Thank you for the thoughtful review and for recognizing the practical significance of the problem as well as the value of a reliability-oriented design.
> We agree that TF-IDF is not semantics-aware and can miss logically important lexical changes such as negation. Our use of TF-IDF is intentionally narrower than semantic equivalence checking. In VeriTrans, the round-trip score is used as a conservative, system-relative consistency signal within a fixed vocabulary and deterministic prompt setting, not as proof that the generated formula is semantically equivalent to the original text. More precisely, our claim is not that TF-IDF validates semantics, but that it functions as a lightweight, deterministic acceptance heuristic for detecting reconstruction drift within a fixed prompt and vocabulary regime. We appreciate that the current wording may not have made this distinction sufficiently prominent, and we would clarify this more explicitly in a revision.
> Regarding semantic metrics such as BERTScore or LLM-as-a-judge, we did consider the broader idea of richer semantic validators. We did not include them in this paper because they would weaken one of the central design goals of VeriTrans: deterministic replay and auditability. Since the paper is framed around reliability-first integration into verification pipelines, we prioritized a lightweight validator that is stable, reproducible, and inexpensive to compute across runs and environments.
> On direct quantitative baselines, this is a fair point. Our hesitation in presenting a direct head-to-head comparison is that systems such as SatLM or Symbolic-CoT are architecturally different: those methods interleave reasoning with solver feedback, whereas VeriTrans externalizes solver interaction as a deterministic post-processing stage and studies fidelity, latency, coverage, and replayability as explicit system properties. Our intended comparison is therefore architectural and systems-oriented rather than a direct one-to-one prompting comparison.
> We also agree that the current paper is limited to propositional logic. This choice is intentional because propositional logic provides a controlled setting for evaluating deterministic integration, validator-gated acceptance, and auditability before extending the framework to richer logics.
> Finally, we agree that a clearer qualitative failure analysis would strengthen the paper. A short discussion of likely failure modes, such as scope or grouping mistakes, omitted implicit constraints, and variable-mapping errors, would improve the presentation and help readers better interpret the remaining error cases.
> Thank you again for the constructive suggestions.

---

### Author Response · Authors · 2026-03-15

We thank all reviewers for their careful reading and constructive feedback. We are encouraged that the reviews recognize both the importance of the problem and the practical value of a reliability-oriented approach.
We would like to clarify the intended scope of the paper. VeriTrans is presented as a reliability-first NL→PL→CNF→SAT pipeline for integrating LLM-based translation into formal-verification workflows. The primary contribution is the combination of constrained NL→PL generation, deterministic symbolic back-end execution, validator-gated acceptance, and replayable artifact logging. The paper is not intended to claim a complete solution to semantic equivalence between natural language and formal logic.
Relatedly, the round-trip score is not meant to serve as a semantics-aware equivalence oracle. Rather, it is used as a conservative consistency signal within a fixed vocabulary, fixed prompt schema, and deterministic reconstruction loop, together with structural checks and thresholded acceptance. We agree that this distinction could have been stated even more explicitly.
We also wish to clarify SatBench's role in our evaluation. In this work, SatBench is used as a corpus of English descriptions paired with gold PL, CNF, and SAT/UNSAT labels. We treat those English descriptions as structured, natural-language, specification-like inputs for evaluating a reliability-oriented translation pipeline. We do not claim that SatBench is an industrial requirements corpus, and we agree that the wording “requirements” was imprecise in places.
Finally, our comparison to prior work is primarily architectural. Solver-in-the-loop systems such as SatLM and Symbolic-CoT have different operating assumptions from VeriTrans, which externalizes solver interaction as a deterministic post-processing stage and evaluates latency, coverage, and replayability as first-class system properties.
We appreciate the comments on writing clarity, metric justification, dataset framing, and qualitative error analysis, and we respond to each reviewer below.